# Irinotecan Induces Disease Remission in Xenograft Mouse Models of Pediatric *MLL*-Rearranged Acute Lymphoblastic Leukemia

**DOI:** 10.3390/biomedicines9070711

**Published:** 2021-06-23

**Authors:** Mark Kerstjens, Patricia Garrido Castro, Sandra S. Pinhanços, Pauline Schneider, Priscilla Wander, Rob Pieters, Ronald W. Stam

**Affiliations:** 1Princess Máxima Center for Pediatric Oncology, 3584 CS Utrecht, The Netherlands; markkerstjens@gmail.com (M.K.); p.garridocastro@gmail.com (P.G.C.); sandra.sofia.m@gmail.com (S.S.P.); p.schneider@prinsesmaximacentrum.nl (P.S.); priswander@hotmail.com (P.W.); r.pieters@prinsesmaximacentrum.nl (R.P.); 2Pediatric Oncology/Hematology, Erasmus MC-Sophia Children’s Hospital, 3015 GD Rotterdam, The Netherlands

**Keywords:** *MLL* translocation, pediatric acute leukemia, irinotecan, xenograft mouse models

## Abstract

Acute lymphoblastic leukemia (ALL) in infants (<1 year of age) remains one of the most aggressive types of childhood hematologic malignancy. The majority (~80%) of infant ALL cases are characterized by chromosomal translocations involving the *MLL* (or *KMT2A*) gene, which confer highly dismal prognoses on current combination chemotherapeutic regimens. Hence, more adequate therapeutic strategies are urgently needed. To expedite clinical transition of potentially effective therapeutics, we here applied a drug repurposing approach by performing in vitro drug screens of (mostly) clinically approved drugs on a variety of human ALL cell line models. Out of 3685 compounds tested, the alkaloid drug Camptothecin (CPT) and its derivatives 10-Hydroxycamtothecin (10-HCPT) and 7-Ethyl-10-hydroxycamtothecin (SN-38: the active metabolite of the drug Irinotecan) appeared most effective at very low nanomolar concentrations in all ALL cell lines, including models of *MLL*-rearranged ALL (*n* = 3). Although the observed in vitro anti-leukemic effects of Camptothecin and its derivatives certainly were not specific to *MLL*-rearranged ALL, we decided to further focus on this highly aggressive type of leukemia. Given that Irinotecan (the pro-drug of SN-38) has been increasingly used for the treatment of various pediatric solid tumors, we specifically chose this agent for further pre-clinical evaluation in pediatric *MLL*-rearranged ALL. Interestingly, shortly after engraftment, Irinotecan completely blocked leukemia expansion in mouse xenografts of a pediatric *MLL*-rearranged ALL cell line, as well as in two patient-derived xenograft (PDX) models of *MLL*-rearranged infant ALL. Also, from a more clinically relevant perspective, Irinotecan monotherapy was able to induce sustainable disease remissions in *MLL*-rearranged ALL xenotransplanted mice burdened with advanced leukemia. Taken together, our data demonstrate that Irinotecan exerts highly potent anti-leukemia effects against pediatric *MLL*-rearranged ALL, and likely against other, more favorable subtypes of childhood ALL as well.

## 1. Introduction

Acute Lymphoblastic Leukemia (ALL) in infants (children <1 year of age) is an aggressive hematologic malignancy characterized by a poor prognosis, with 5-year event-free survival (EFS) rates of ~50% [1,2,3]. In contrast, over the past decades the 5-year EFS for children older than 1 year of age diagnosed with ALL has progressively improved towards ~90%, due to better risk stratification and accordingly adjusted treatment protocols [4,5]. Furthermore, approximately 80% of infant ALL cases carry leukemia-specific chromosomal translocations of the *Mixed Lineage Leukemia* (*MLL*, or *KMT2A*) gene, and this patient group in particular fares significantly worse with EFS rates of 30–40% [1,2]. Evidently, there is a dire need for improved therapeutic strategies to ameliorate clinical outcomes for these very young patients.

*MLL* translocations give rise to chimeric MLL fusion proteins, which induce inappropriate histone modifications by recruitment of the histone methyl transferase DOT1L [6,7]. This induces a perturbed epigenetic landscape resulting in severely altered gene expression signatures and DNA methylation patterns, giving rise to a type of leukemia which both biologically and clinically significantly differs from pediatric ALL without *MLL* translocations [8,9,10,11,12]. We and others have been investigating therapeutic approaches targeting components of the epigenetic machinery important in *MLL*-rearranged ALL. So far, this has led to the discovery of inhibitors against DOT1L (e.g., EPZ004777), DNA methyltransferases (e.g., 5-azacytidine or zebularine), BET family proteins (e.g., I-BET151) and histone deacetylases (HDACs) (e.g., vorinostat or panobinostat) as potential drug candidates [9,13,14,15,16,17].

Although these epigenetic-based drugs do show promising results, the transition from preclinical studies towards clinical applications unfortunately remains an elaborate and time-consuming process. Therefore, awaiting clinical evaluation of these inhibitors, we decided to adopt a drug repurposing approach using drug library screening of >3500 FDA-approved and off-patent therapeutic agents. Using this strategy, we aimed to identify effective therapeutics against *MLL*-rearranged ALL, which have been characterized and approved for other human diseases and thus expediting their potential transition into clinic [18].

In this study we identified the camptothecin-derivative 7-Ethyl-10-hydroxycamptothecin (SN-38), and in particular its pro-drug irinotecan (Camptosar), as highly effective agents against *MLL*-rearranged ALL. In fact, we here demonstrate that irinotecan monotherapy successfully induced disease remission in xenograft mouse models of *MLL*-rearranged ALL.

## 2. Materials and Methods

### 2.1. Cell Culture

The *MLL*-rearranged ALL cell lines SEM (MLL-AF4^+^) and KOPN8 (MLL-ENL^+^), the BCP-ALL cell lines REH (TEL-AML1^+^), 697 (E2A-PBX^+^) and Sup-B15 (BCR-ABL^+^), and the T-ALL cell line Jurkat (pseudodiploid) were obtained from the German Collection of Microorganisms and Cell Cultures (DSMZ, Braunschweig, Germany). The *MLL*-rearranged ALL cell line RS4;11 (MLL-AF4^+^) was obtained from The Global Biosource Center (ATCC, Middlesex, UK). Leukemia cell lines were cultured in RPMI-1640 with GlutaMAX, 10% Fetal Calf Serum, 100 IU/mL penicillin, 100 IU/mL streptomycin and 0.125 µg/mL amphotericin B (Invitrogen Life Technologies, Waltham, MA, USA) at 37 °C under 5% CO_2_ atmosphere. Regular DNA fingerprinting and mycoplasma testing were performed to ensure cell line integrity. Primary samples derived from *MLL*-rearranged infant ALL patients were obtained at the Sophia Children’s Hospital (Rotterdam, the Netherlands) as part of the international INTERFANT treatment protocol. Informed consent was obtained according to the Declaration of Helsinki. The study was approved by the Ethics Committee of all collaborating institutions and registered with the National Cancer Institute (ClinicalTrials.gov identifier: NTC0550992) and with the European Clinical Trials database (EudraCT 2005-004599-19) (24-11-2005). Processing of samples occurred as described previously [19,20]. Leukemic blast percentage was at least 90%, as confirmed by May–Grünwald–Giemsa counterstained cytospins.

### 2.2. Drug Library Screening

We purchased the drug libraries ‘Spectrum Collection’ (MicroSource, Gaylordsville, CT, USA) and ‘Prestwick library’ (Prestwick Chemical, Illkirch, France), and the anti-neoplastic ‘Sequoia library’ (Sequoia Research Products, Pangbourne, UK). DMSO dissolved stocks were diluted in non-supplemented RPMI, and cytotoxicity was assessed using MTT/MTS assays (DMSO concentration <0.5%). Data were normalized to vehicle control, and cell viability measures of 100% or more were deemed completely viable and unaffected by drug treatment. Heatmaps were generated using Gene-E software (Broad Institute, Cambridge, MA, USA).

### 2.3. In Vitro Drug Exposures

Cell proliferation was tracked by propidium-iodide exclusion on a MACSQuant flow cytometer (Miltenyi). Apoptosis was assessed using flow cytometry with the PE Annexin-V Apoptosis Detection Kit (BD Pharmingen, San Diego, CA, USA). Flow cytometry data were analyzed using FlowJo software (version 10.7: BD Biosciences, Franklin Lakes, NJ, USA). For whole cell lysates, cells were washed with PBS and lysed on ice in RIPA buffer supplemented with protease and phosphatase inhibitors. MTS and MTT dose–response data were acquired in duplicate and presented as mean +/− s.e.m. (cell lines) or mean +/− SD (patient samples).

### 2.4. Western Blot

Protein lysates were resolved on pre-cast SDS-polyacrylamide gels (TGX, Bio-Rad, Veenendaal, The Netherlands) and transferred to nitrocellulose membranes using the Transblot Turbo Transfer System (BioRad, Veenendaal, The Netherlands). Membranes were blocked with 5% BSA or skim milk in TBS and probed with primary antibodies against PARP, (phospho-)Chk2, (phospho-)H2AX (Cell Signalling Technologies) or β-actin (Abcam, Cambridge, UK), and subsequently with fluorophore-conjugated secondary antibodies. Images were acquired using the Odyssey imaging system (LI-COR, Leusden, The Netherlands).

### 2.5. Animal Experiments

Animal experiments were performed according to Dutch legislation and approved (10-12-2014) by the Erasmus MC Animal Ethical Committee, Rotterdam, The Netherlands (EMC3389). Briefly, immunodeficient NSG mice were transplanted with SEM-SLIEW or patient-derived leukemic cells, and leukemia progression was assessed through intra-vital imaging or human CD45^+^ cell counts in peripheral blood samples. Vehicle or irinotecan (40 mg/kg) treatment was administered intraperitoneally three times per week. Mice were humanely euthanized and tissue samples were acquired for further analysis. Statistical tests were performed in GraphPad Prism. Spleen weight differences were assessed using non-parametric tests (i.e., Kruskal–Wallis or Mann–Whitney). Differences in leukemic cell infiltration from FACS data were assessed using parametric tests (i.e., one-way ANOVA and unpaired *t*-test).

## 3. Results

### 3.1. Drug Library Screening Identifies Camptothecin Derivatives as Promising Leads

We performed drug library screens using the Spectrum and Prestwick drug libraries (consisting of 2320 and 1200 therapeutically diverse but mainly FDA-approved compounds, respectively) at a drug concentration of 1 µM on various *MLL*-rearranged ALL cell lines as well as non-*MLL* B-cell precursor (BCP) ALL cell lines. While the majority of drugs did not affect the leukemia cell lines tested, approximately 12% of the compounds inhibited *MLL*-rearranged ALL cell viability by at least 20%, for both the Spectrum (Figure 1a) and Prestwick (Figure 1b) libraries. Comparing the 200 most effective Spectrum library compounds for the *MLL*-rearranged ALL cell lines with those for the BCP-ALL cell lines, revealed an overlap of 148 compounds (74%), suggesting that most of these drugs are not specifically targeting *MLL*-rearranged ALL (Figure 1c,d). Moreover, the 52 non-overlapping compounds more specifically affecting *MLL*-rearranged ALL cells, mostly were effective in only one of the three *MLL*-rearranged ALL cell lines, i.e., RS4;11 (Appendix A; top). Similar results were obtained for the 100 most effective Prestwick library compounds, with an overlap of 82 (82%) and with the 18 non-overlapping compounds specific for *MLL*-rearranged ALL only being effective in one cell line, being either RS4;11 or KOPN8 (Appendix A; bottom). Among the most effective drugs we found vincristine, cytarabine and vorinostat, which either are already being used in current treatment protocols or have been identified previously as therapeutic options for *MLL*-rearranged ALL, confirming the validity of our screening approach.

Interestingly, we also found various corticosteroid drugs to be more effective in the *MLL*-rearranged ALL cell lines compared to the BCP-ALL cell lines, which was probably due to the BCP-ALL cell line REH being non-responsive to glucocorticoids as it lacks detectable expression of the glucocorticoid receptor (GR). Nonetheless, we tested several of the corticosteroid drugs on the *MLL*-rearranged ALL cell lines SEM and KOPN8 but found none to be more potent or effective than prednisolone (Appendix A). Therefore, this specific class of drugs was excluded from further investigation.

Since several drugs completely diminished ALL cell viability, we asked whether lower drug concentrations could reveal leukemia subtype specificity that was missed at the 1 µM concentration. Therefore, 54 potential leads were selected based on their inhibitory effect on *MLL*-rearranged ALL cells, and further validated at 100 nM and 10 nM concentrations, yielding a narrow selection of very potent compounds (Appendix A). However, none of these drugs appeared to display leukemia subtype specificity.

In addition, the anti-neoplastic Sequoia drug library (consisting of 165 anti-neoplastic chemotherapeutics commonly used in the treatment of human cancers) was screened in the same setup at 1 µM drug concentration. This drug screen yielded a substantial number of effective compounds, i.e., 52 compounds with >75% inhibition of *MLL*-rearranged ALL cell viability (Figure 1e; left). Further examination of efficacy at 100 nM and 10 nM concentrations identified a select group of highly potent agents (Figure 1e; center and right).

Figure 1f,g show the effects on cell viability in *MLL*-rearranged ALL and BCP-ALL cell lines for the top five drugs from the Spectrum and Prestwick libraries combined, and the top five drugs from the Sequoia library at a 10 nM drug concentration. Among these drugs, gemcitabine appeared to be the only chemotherapeutic more specifically targeting *MLL*-rearranged ALL cells compared to BCP-ALL cells. Although no other *MLL*-rearranged ALL specific agents could be identified, the top compounds in all three drug libraries appeared to be camptothecin and its derivatives 10-hydroxycamptothecin (10-HCPT), 7-ethyl-10-hydroxycamptothecin (SN-38) and topotecan (Figure 1f,g). For additional validation, we tested whether the 54 selected Spectrum/Prestwick drugs as well as all agents in the Sequoia library inhibited a primary *MLL*-rearranged ALL patient sample to a similar extent as was observed for the ALL cell lines. While some of the Spectrum/Prestwick drugs performed differently on the primary patient samples compared to the cell line models, camptothecin and its derivatives remained highly effective (Figure 1h).

### 3.2. Camptothecin Derivative SN-38 Most Potently Inhibits ALL Cell Viability

To validate the results from our drug library screens, we tested camptothecin and its derivatives SN-38 and 10-HCPT using dose–response curves on the *MLL*-rearranged and BCP-ALL cell lines used in the original drug library screens, and additionally included the T-ALL cell line Jurkat. All three agents strongly inhibited leukemic cell viability in each cell line tested, with IC50 values in low nanomolar ranges (Figure 2a–c). Next, the efficacy of the most potent agent, SN-38, was further validated in multiple primary *MLL*-rearranged infant ALL patient samples (Figure 2d). The IC50 values of SN-38 in the cell line models (ranging from 1.4 to 5.6 nM) was notably lower than in the primary patient samples (ranging between 13.9 and 434 nM), possibly due to the fact that SN-38 activity requires cell proliferation, and patient-derived leukemic cells hardly divide in vitro. Hence, patient-derived xenograft (PDX) mouse models are required to explore the actual efficacy of this drug against primary MLL-rearranged infant ALL in vivo (see below).

### 3.3. SN-38 Induces DNA Damage and Apoptotic Cell Death in ALL

To elucidate the mechanism underlying SN-38 mediated inhibition of cell viability, we exposed the *MLL*-rearranged ALL cell line SEM to 5 nM and 25 nM of SN-38 and assessed changes in proliferation over time. We observed reduced cell numbers at both concentrations already after 24 h, which progressively decreased after 48 and 72 h (Figure 3A). Cell death determination by flow cytometry after 8, 24 and 48 h showed that apoptosis was initiated already after 8 h of exposure with 25 nM of SN-38 and progressed dramatically at both concentrations after 48 h (Figure 3B). Apoptosis induction was further confirmed by a clear detection of PARP cleavage (Figure 3C).

During chromatin replication or transcription, topoisomerase I (TOP1) induces DNA single strand breaks to release DNA supercoiling-induced torsional stress [21] SN-38 binds to TOP1 to prevent DNA re-ligation, thereby locking TOP1 onto the DNA. The resulting DNA/TOP1/SN-38 complexes can lead to accumulation of replication fork collision-induced DNA breaks, especially in fast-dividing cells. Important hallmarks of camptothecin-derivative mediated DNA breaks are enhanced phosphorylation of the DNA damage response (DDR) proteins CHK2 and H2AX (γH2AX) [22,23]. We observed increased phosphorylation of CHK2 and H2AX, as well as slightly increased total H2AX protein levels, after exposing SEM cells to 25 nM SN-38, as determined by immunoblotting (Figure 3C). Similar results were obtained for the BCP-ALL cell line 697 (Appendix A).

### 3.4. The SN-38 Pro-Drug Irinotecan Effectively Inhibits MLL-Rearranged ALL In Vivo

To assess the efficacy of SN-38 in vivo, we used a previously established *MLL*-rearranged ALL xenograft mouse model [15]. However, since SN-38 is notorious for its poor bioavailability, we instead decided to treat our mice with irinotecan (CPT-11, or Camptosar), which is a pro-drug metabolized into SN-38 in vivo by native carboxylesterases [22,23]. Immunodeficient NSG mice (*n* = 19) were xenotransplanted with our *MLL*-rearranged ALL luciferase reporter cell line SEM-SLIEW and equally allocated over vehicle (*n* = 10) and irinotecan (*n* = 9; 40 mg/kg) treatment groups, ensuring comparable leukemia burden before treatment (Appendix A). Irinotecan or vehicle was administered 3 times per week via intraperitoneal injections. Interestingly, after 14 and 20 days, bioluminescent imaging revealed clear luminescence in vehicle-treated mice (indicative of progressing leukemia), while no luminescence was observed in irinotecan-treated mice (Figure 4A). As relapses in *MLL*-rearranged ALL patients often occur early during treatment [1,2], suggesting that small subsets of leukemic cells evade therapy [24], we studied whether the observed ablation of leukemia by irinotecan would be maintained after cessation of irinotecan treatment. Therefore, a subset (*n* = 6) of the irinotecan-treated mice were sacrificed at day 28 and used to investigate leukemic infiltration of tissues, while the remaining *n* = 3 irinotecan-treated mice were kept alive without any further treatment (from here on referred to as the treatment follow-up group) (Figure 4A; right), while disease monitoring continued by weekly bioluminescent imaging. Interestingly, the intra-vital images showed no outgrowth of leukemia for up to 42 days off treatment (day 70) in these mice (Figure 4B), indicating no signs of leukemia relapse since cessation of irinotecan administrations. Quantification of the total flux (emitted photons/second) showed a rapid increase in bioluminescence in the vehicle-treated mice, marking progressive leukemia, whereas we observed a lack of signal for mice on the irinotecan treatment arm, which persisted after the end of treatment in the follow-up group, indicating no detectable leukemia relapse (Figure 4C).

After sacrificing mice from the vehicle (*n* = 6), irinotecan (*n* = 6) and follow-up (*n* = 3) groups, relevant tissues were extracted and further analyzed. In addition, we included two healthy, non-transplanted and untreated NSG mice as basal controls. As expected, vehicle-treated control mice characteristically presented with splenomegaly, while irinotecan-treated mice, both in the initial treatment and the follow-up group, had significantly lower spleen weights (*p* < 0.0001), resembling the spleens of healthy control mice (Figure 4D,E). Next, we processed the spleens, bone marrow and peripheral blood and determined leukemic cell burden in these tissues using multicolor flow cytometry, as outlined in Appendix A. In the homogenized spleens from the vehicle group, approximately 45% CD19^+^ human leukemia cells were observed, whereas no leukemic cells were detected in spleens from irinotecan-treated mice or the treatment follow-up group (Figure 4F). Similarly, bone marrow and peripheral blood samples of vehicle-treated mice on average contained ~85% and ~4% of human leukemic cells, respectively, whereas no leukemic cells were detected in the bone marrow and peripheral blood of the irinotecan-treated or follow-up groups (Figure 4G,H).

### 3.5. Irinotecan Cures Mice with Advanced MLL-Rearranged ALL

While irinotecan monotherapy was able to completely block human *MLL*-rearranged ALL expansion in mice when treatment was initiated shortly after xenotransplantation, this may not represent a clinically relevant model, as patients present with full-blown leukemia at diagnosis. Therefore, we tested whether irinotecan could also cure advanced leukemia in xenotransplanted mice. Four mice from the vehicle-treated group (as presented in Figure 4A) with advanced leukemia were selected for subsequent treatment with irinotecan (40 mg/kg) (from here on referred to as curative group), which was initiated at 16 days after transplantation. Remarkably, already after 2 injections of irinotecan (in the first week of treatment), intra-vital bioluminescent imaging showed the systemic leukemia starting to regress, which was clearly visible in all mice (Figure 5A; 20 days). Moreover, successive treatment further diminished the leukemia considerably, and although 1 of these mice was found dead (day 26; unknown cause), quantification of the bioluminescence from the remaining *n* = 3 mice confirmed a rapid decline in leukemic burden (Figure 5B). Interestingly, at the end of this curative study (day 36), the spleens from the curative group were significantly smaller than those from the vehicle-treated group (Figure 5C). Furthermore, no human leukemic cells were detected in the spleens, bone marrows or peripheral blood of the curatively treated mice (Figure 5D–F; respectively).

Additionally, histological staining revealed the absence of leukemic cell infiltration in tissues from irinotecan-treated mice, resembling healthy tissues (Appendix A). Of interest is the observation that brain tissues of all the irinotecan-treated mice, and particularly the mice in the curative group, were devoid of leukemic cells, while we observed leukemic infiltration into the leptomeningeal space in mice of the vehicle group (Appendix A). This is of great clinical relevance, as it indicates the ability of irinotecan to eradicate central nervous system (CNS) infiltration; CNS leukemia involvement is common in *MLL*-rearranged infant ALL and represents an adverse prognostic factor.

### 3.6. Irinotecan Shows Potent Anti-Leukemic Effects in MLL-Rearranged ALL Patient-Derived Xenograft Mouse Models

To exclude the possibility that the observed results of the irinotecan treatment are only valid in leukemia cell line models, we also investigated the anti-leukemic efficacy of irinotecan in patient-derived xenograft (PDX) models with leukemic cells from two *MLL*-rearranged infant ALL patients (UPID VU9815, corresponding to patient C in Figure 1e, and UPID 788). After transplantation, disease development was monitored by tail vein bleeding to determine presence of human CD45^+^ leukemic cells in the peripheral blood. Treatment was initiated when >1% CD45^+^ cells were detected in the peripheral blood as determined by multi-color flow cytometry. One PDX model (UPID VU9815) already displayed more substantial engraftment levels at treatment initiation, with 7–44% human CD45^+^ leukemic cells. Mice were randomly allocated to either the vehicle control group (*n* = 4 UPID VU9815, *n* = 4 UPID 788) or the irinotecan treatment group (*n* = 4 UPID VU9815, *n* = 4 UPID 788). The treatment regimen was comparable to the cell line xenografts. Regardless of the PDX model, in less than two weeks after treatment initiation a substantial decrease in human chimerism was observed in the irinotecan treatment mice, while an exponentially increased accumulation of human CD45^+^ leukemic cells was observed in the vehicle groups. After 10 doses of irinotecan, mice were in complete remission (<0.2%), while the leukemic burden in control mice was 38–83% (Figure 6A). Both vehicle and irinotecan-treated mice were sacrificed, and tissues were harvested and analyzed. As observed in the cell line xenografts, human leukemic cells were almost completely eradicated in the bone marrow, peripheral blood and spleen of irinotecan-treated PDX mice (Figure 6B). In line with this, spleens of irinotecan-treated mice were significantly smaller as compared with spleens from the vehicle group (Figure 6C).

In parallel to the curative set-up, we also established a separate treatment follow-up arm from both PDX models (VU9815, UPID788) to monitor for relapse of the disease, similar to the treatment arms in the cell line model. Irinotecan treatment was stopped after 10 dosages, and the irinotecan-treated mice were kept alive in the treatment follow-up group, whereas the control mice were sacrificed. Here, in contrast to the cell line model, the irinotecan-treated mice of the follow-up group relapsed after 4 weeks following the last irinotecan administration (Figure 7A). The mice were sacrificed, and the tissues analyzed, showing systemic dissemination of human leukemic cells (Figure 7B,C). A possible explanation for these relapses may well be the sporadic presence of human CD45^+^ leukemic cells in the bone marrow and spleen at the end of treatment, which could act as a cell reservoir inducing leukemia relapse.

## 4. Discussion

*MLL*-rearranged infant ALL is an aggressive hematologic malignancy with a poor prognosis, which urgently requires improved therapeutic strategies [1,2]. In this study we employed a drug repurposing approach by screening different drug libraries consisting of a total of 3685 mostly FDA-approved drugs, adopting the rationale that such therapeutics could be expeditiously transitioned to clinical practice. We successfully identified the camptothecin derivative SN-38, and its pro-drug irinotecan, as a very promising therapeutic option for *MLL*-rearranged infant ALL. Our data, combined with the fact that irinotecan is a well-characterized anti-neoplastic drug, strongly advocate for clinical evaluations of irinotecan for the treatment of *MLL*-rearranged infant ALL.

Although the rationale behind drug repurposing through screening of FDA-approved drug libraries in itself is not innovative, it has not previously been reported for *MLL*-rearranged ALL with drug libraries of this magnitude. However, earlier drug discovery attempts involving FDA-approved drug screens have resulted in the identification of leukemia-subtype specific inhibitors against for example AML1-ETO-positive acute myeloid leukemia (AML) and NOTCH1-mutated T-cell ALL [25,26].

Our drug library screens resulted in the identification of the camptothecin-derived drug class of TOP1 inhibitors, which effectively inhibited *MLL*-rearranged ALL cell lines, as well as other ALL subtypes, with SN-38 as the most potent inhibitor of ALL cell viability. The mechanism of action for this drug class involves the generation of DNA/TOP1/drug complexes, which lead to collisions with either DNA synthesis- or transcriptional elongation-related replication forks, thereby inducing double-strand DNA breaks [27,28]. We observed higher SN-38 IC_50_ values for infant ALL patient samples compared to the ALL cell line models, which might be explained by the limited division capacity of patient-derived leukemic cells in vitro and the high proliferation rate of immortalized cell lines. Still, nanomolar SN-38 concentrations could effectively inhibit primary leukemia cells in vitro, even in the absence of leukemic cell proliferation. The DNA breaks induced by SN-38 activate DNA damage response pathways, including the cell cycle regulating ATM-Chk2 pathway, and this is accompanied by the formation of phosphorylated H2AX (γ-H2AX) foci [29,30]. We confirmed that low nanomolar concentrations of SN-38 induced the DNA-damage response pathway in leukemic cells via enhanced phosphorylation of Chk2 and H2AX, followed by rapid induction of apoptosis.

The anti-leukemic effects of SN-38 could be validated in *MLL*-rearranged ALL xenograft models, where the SN-38 pro-drug irinotecan effectively inhibited leukemia development. Moreover, since the leukemia is usually fully disseminated in patients at diagnosis, we started administering irinotecan to mice with advanced leukemia to mimic the start of treatment in a clinical setting. In all mice, progressive leukemia drastically regressed within days, with no detectable residual leukemic cells at the endpoint of the experiment in cell line xenografts, and minimal residual disease levels in PDX mice. These very low levels of residual human leukemic cells most likely represented the relapse-initiating cells in the PDX follow-up treatment arm. Prolonging the treatment should help to eradicate this, as it has been previously described that dosage regimen and duration is pivotal for the efficacy of camptothecin-derivatives [31], and the initial irinotecan treatment duration of the PDX mice was shorter than the treatment of the cell line xenografts due to the exponential disease kinetics of the vehicle control group. Nevertheless, it is important to emphasize that while numerous studies show inhibition of leukemic expansion in xenograft mouse models when treatment commences shortly after xenotransplantation, complete regression of advanced leukemia in mice, especially by single-agent therapy, has only been reported sporadically [32,33,34,35,36,37]. Interestingly, Jones et al. recently reviewed results from a pediatric ALL xenograft platform, including one *MLL*-rearranged acute leukemia sample, and reported TOP1 inhibitor topotecan effectively inhibited ALL in seven out of eight PDX models [38]. Additionally, a recent drug profiling study on ex vivo PDX material showed topotecan inhibited three out of four *MLL*-rearranged BCP-ALL samples with IC_50_ values in the order of 10–100 nM [39]. Moreover, clinical trials investigating TOP1 inhibitors for acute leukemia have been reported, showing promising results [40,41,42,43,44]. This could raise the question why these inhibitors have not progressed into clinical application. Possibly, the improved outcome with conventional chemotherapy for pediatric ALL in the past decades has favored implementation of targeted therapies over chemotherapeutics like TOP1 inhibitors. In contrast, outcome in *MLL*-rearranged infant ALL has stagnated over the last decades and novel treatment rationales are urgently required; the results from the present study, particularly in the context of the clinical data on irinotecan treatment in literature, suggest that TOP1 inhibitors may indeed be a valuable addition to infant ALL treatment protocols.

Advantageously, irinotecan has been on the market for two decades already, and the amount of available information and clinical data is substantial. Besides the extensive use in adult patients for the treatment of several solid tumors, irinotecan has been an important component of pediatric sarcoma therapy for the past 15 years, and earlier clinical experiences and challenges have recently been reviewed, including important considerations for dosage regimens [45]. Additionally, Thompson et al. previously established a pharmacokinetics model for irinotecan in non-infant pediatric cancer patients and concluded metabolism and SN-38 plasma levels were dependent on age [46]. These studies should aid and expedite the transition of irinotecan/SN-38 towards clinical application for infants with *MLL*-rearranged ALL, and possibly other high-risk types of childhood leukemia.

Moreover, whereas irinotecan/SN-38 monotherapy already shows promise, combination therapy with HDAC inhibitors has been shown to synergistically inhibit other malignancies [47,48,49,50,51]. Together with our previous work, demonstrating that HDAC inhibition effectively and specifically targets *MLL*-rearranged ALL both in vitro and in vivo [15], this opens further treatment possibilities for infant ALL, such as irinotecan and HDAC inhibitor combination therapy, potentially reducing dosages and/or alleviating treatment regimens.

In summary, the data presented here strongly advocate for the implementation of irinotecan (or comparable compounds) in the treatment of *MLL*-rearranged infant ALL.

## Figures and Tables

**Figure 1 biomedicines-09-00711-f001:**
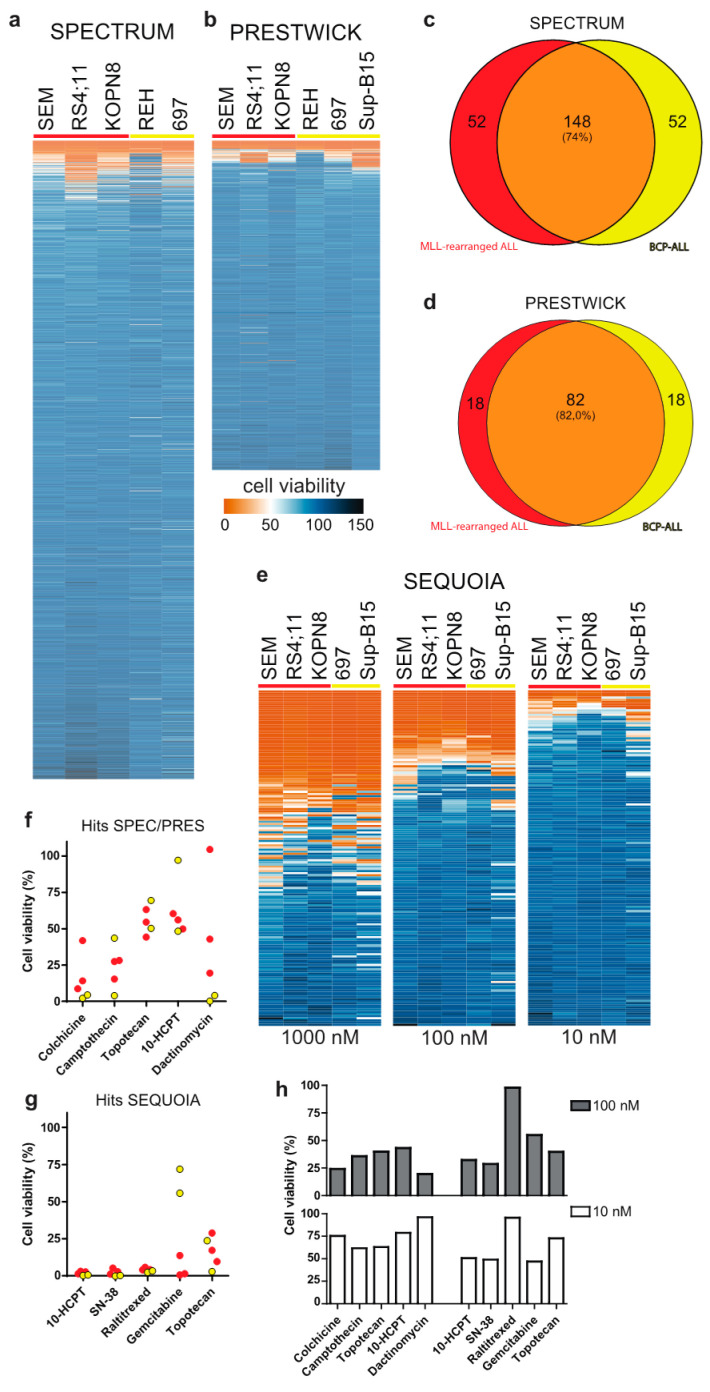
Drug library screens: (**a**,**b**) Heatmaps of ALL cell viability after exposure to the Spectrum and Prestwick drug libraries (1 µM, 96 h), respectively. (**c**,**d**) Venn-diagram representation of the top 200 and top 100 screening hits from the Spectrum and Prestwick libraries, respectively, against *MLL*-rearranged ALL (red) and BCP-ALL (yellow). (**e**) Heat maps showing the effect of the anti-neoplastic Sequoia library against ALL cell viability (1 µM, 100 nM and 10 nM; left, middle and right, respectively; 96 h). (**f**,**g**) Cell viability data for the 5 most effective Spectrum and Prestwick library derived hits and the most effective Sequoia library hits, respectively (10 nM, 96 h). *MLL*-rearranged ALL cell lines are shown in red, BCP-ALL cell lines are shown in yellow. (**h**) Cell viability of a primary *MLL*-rearranged infant ALL sample exposed to the top hits at 100 nM (top; grey bars) and 10 nM (bottom; white bars).

**Figure 2 biomedicines-09-00711-f002:**
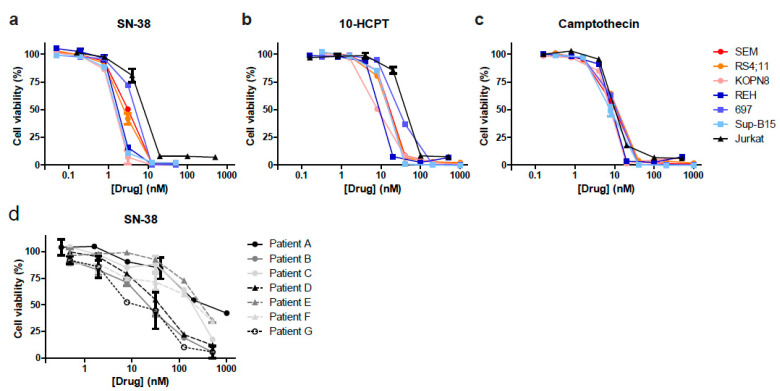
SN-38 is the most potent camptothecin derived inhibitor of ALL in vitro. (**a**–**c**) Dose–response curves for the selected hits SN-38 (7-ethyl-10-hydroxycamptothecin), 10-HCPT (10-hydroxycamptothecin) and camptothecin against *MLL*-rearranged ALL cell lines SEM, KOPN8 and RS4;11 (red colors), BCP-ALL cell lines REH, 697 and Sup-B15 (blue colors), and T-ALL cell line Jurkat (black). (**d**) Dose–response curves showing cell viability data of *n* = 7 primary *MLL*-rearranged infant ALL samples exposed to SN-38.

**Figure 3 biomedicines-09-00711-f003:**
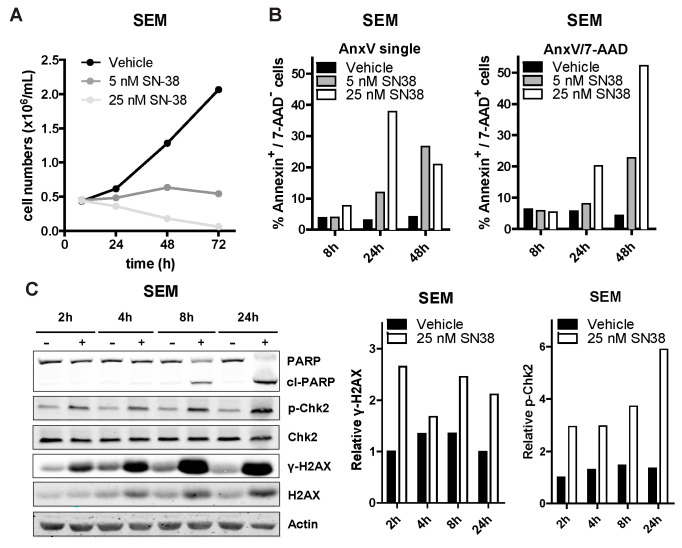
SN-38 induces DNA damage and apoptosis in ALL cells. (**A**) SEM cell counts over time (8 h, 24 h, 48 h and 72 h) after exposure to vehicle (DMSO), 5 nM or 25 nM of SN-38 (black, dark grey and light grey lines; respectively; *n* = 3). (**B**) Percentages of (early and late) apoptotic SEM cells exposed to SN-38. Representative of three separate experiments. (**C**) Western blots of SEM lysates (left) after exposure to vehicle (-) or 25 nM SN-38 (+) for the indicated time-points. Quantification of the phosphorylated H2AX (γ-H2AX) level relative to total H2AX and relative Chk2 phosphorylation in the SEM samples is shown in the center and right graphs, respectively.

**Figure 4 biomedicines-09-00711-f004:**
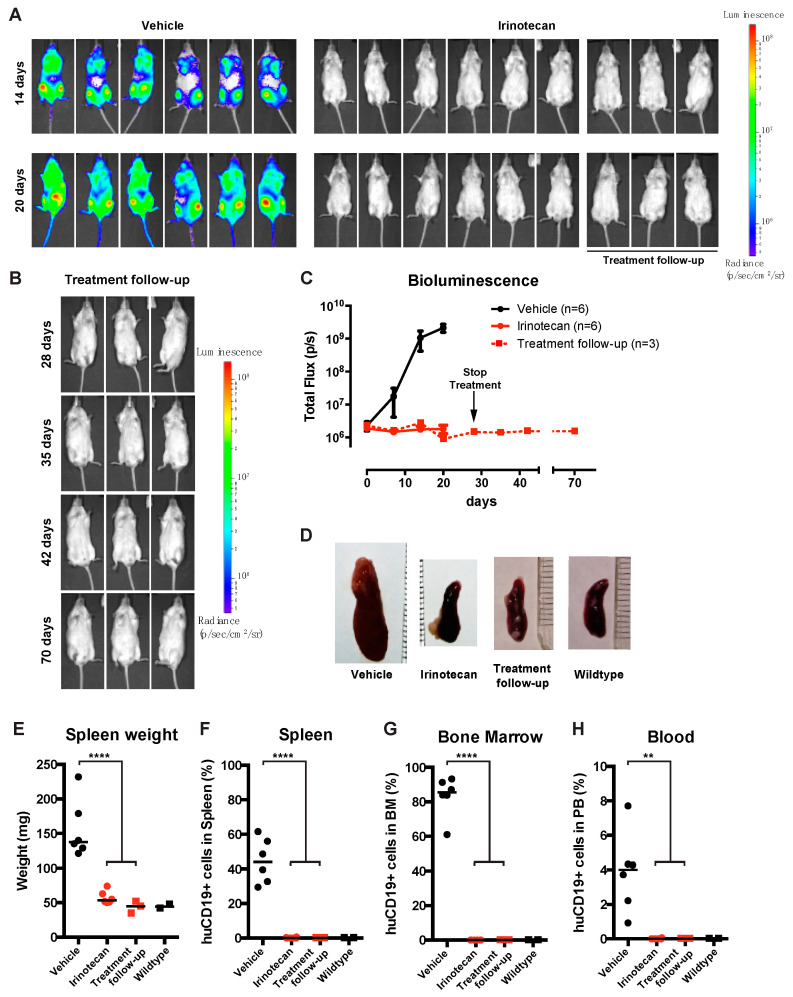
SN-38 pro-drug irinotecan can impede *MLL*-rearranged leukemia outgrowth in vivo. (**A**) Intra-vital imaging of SEM-SLIEW transplanted mice treated with either vehicle (*n* = 6; left) or 40 mg/kg irinotecan (*n* = 9; right) after 14 or 20 days (top and bottom, respectively). All images were generated with the same luminescence scale (legend on far-right side). (**B**) Intra-vital imaging of 3 irinotecan treated mice after cessation of treatment on day 28 (prolonged remission group), tracked until day 70. Luminescence scale is the same as in (**A**). (**C**) Quantification of bioluminescent signal from the vehicle (black), irinotecan (red circles, solid line) and prolonged remission (red squares, dashed line) mice. The black arrow indicates cessation of irinotecan treatment (prolonged remission group; day 28). Data presented as median total flux (emitted photons/second) +/− sd. (**D**) Representative images of mouse spleens from the vehicle, irinotecan, and prolonged remission groups, as well as a wildtype mouse spleen. (**E**) Individual spleen weights for vehicle (black circles), irinotecan (red circles) treatment groups, and prolonged remission (red squares) and wildtype (black squares) mice. Horizontal bars indicate median. (**F**–**H**) Percentage live human CD19-positive cells derived from homogenized spleen, bone marrow and peripheral blood of vehicle (black circles), irinotecan (red circles), prolonged remission (red squares) and wildtype (black squares) mice. Horizontal bars indicate median. ** *p* < 0.05, **** *p* < 0.0001.

**Figure 5 biomedicines-09-00711-f005:**
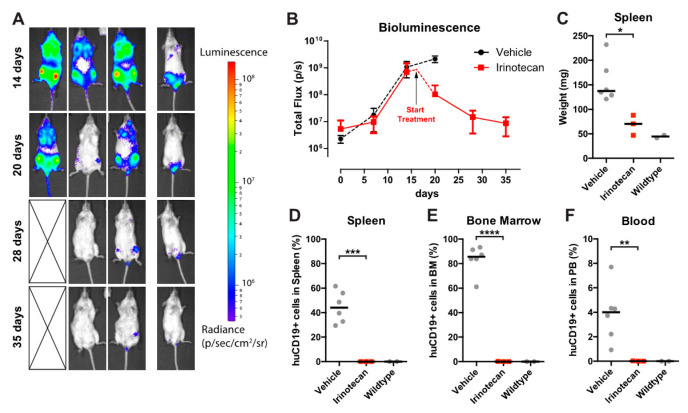
Irinotecan treatment cures mice with advanced leukemia. (**A**) Intra-vital imaging of SEM-SLIEW transplanted mice with advanced leukemia (14 days; top row) and when started on curative irinotecan treatment (20, 28 and 34 days). Deceased mouse is indicated by cross. Luminescent scale is identical to Figure 4A,B. (**B**) Quantification of bioluminescence from the curative group (*n* = 3; deceased mouse excluded). Vehicle bioluminescent signal (black dotted line) from Figure 4 included for reference. The black arrow indicates initiation of curative irinotecan treatment (day 16). (**C**) Spleen weight of the curative irinotecan treated mice (red squares). Vehicle (left grey circles) and wildtype (right grey circles) data from Figure 4 included for comparison. Horizontal bars represent median values. (**D**–**F**) Percentage of live human CD19-positive cells in spleen, bone marrow and peripheral blood (respectively) of curative group (red squares). Again, vehicle (left grey circles) and wildtype (right grey circles) data from Figure 4 are included. Horizontal bars represent group medians. * 0.01 < *p* < 0.05; ** 0.001 < *p* < 0.01; *** 0.0001 < *p* < 0.001; **** *p* < 0.0001.

**Figure 6 biomedicines-09-00711-f006:**
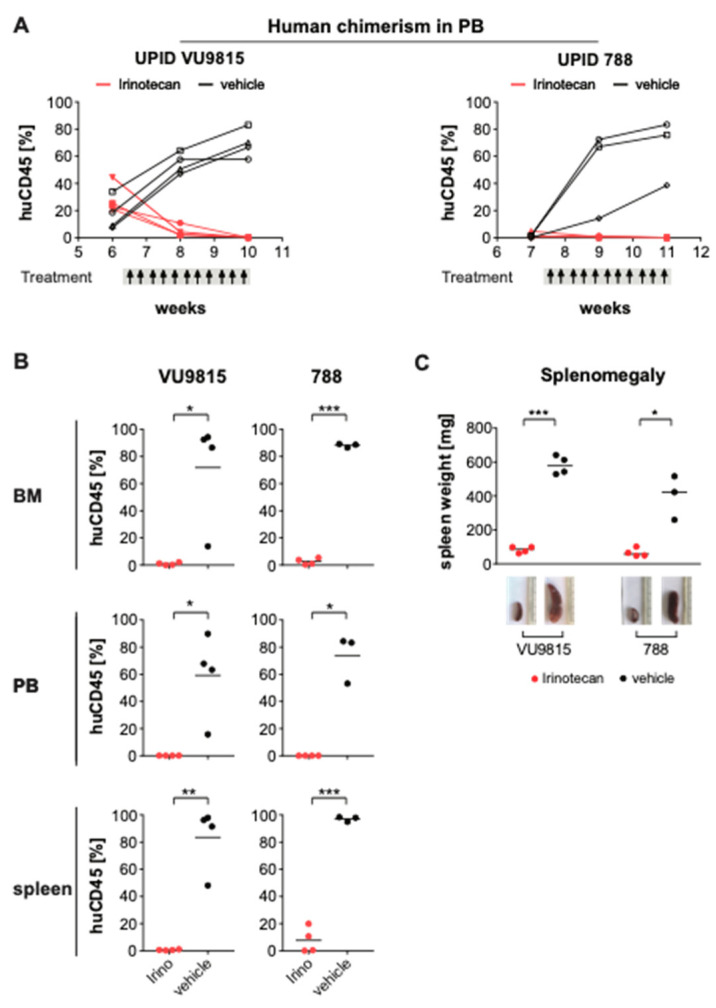
Irinotecan blocks leukemia progression and induces remission in *MLL*-rearranged ALL patient-derived xenografts. (**A**) Human chimerism in the peripheral blood (PB) of two patient-derived xenograft (PDX) mouse models (UPID VU9815, UPID788) was monitored over time by multi-color flow cytometry. Presence of >1% human CD45+ human ALL cells was the threshold to initiate treatment with vehicle or irinotecan (40 mg/kg). Per PDX model, *n* = 4 mice were randomly allocated to each treatment arm. Administration occurred i.p. three times per week, for a total of 10 dosages. Changes in human leukemic cell burden were monitored every other week. One mouse in the UPID VU9815 was excluded due to technical reasons (i.e., died during bleeding procedure). (**B**) At the end of treatment, mice were euthanized and human leukemic cell infiltration in bone marrow (BM), PB and spleen measured by multi-color flow cytometry. Statistical differences were determined using Student’s *t*-test with Welch correction; * = *p* < 0.05; ** = *p* < 0.01; *** = *p* < 0.001. (**C**) Splenomegaly was illustrated by differences in spleen weights and sizes between irinotecan- and vehicle-treated xenograft mice. Statistically significant differences were analyzed using Mann–Whitney U testing; * = *p* < 0.05; *** = *p* < 0.001.

**Figure 7 biomedicines-09-00711-f007:**
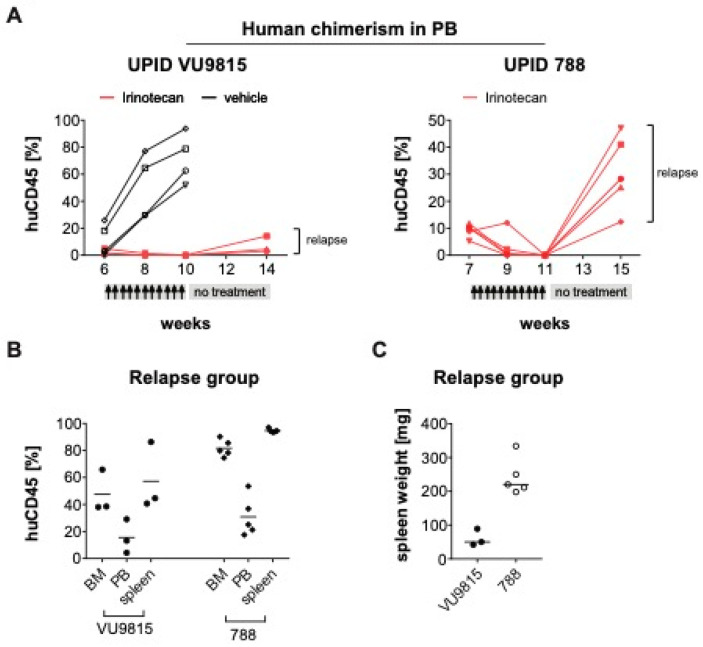
Minimal residual disease causes relapse in *MLL*-rearranged ALL PDX models after treatment stop (**A**). Human chimerism in the peripheral blood (PB) of two PDX mouse models (UPID VU9815, UPID 788) was monitored over time by multi-color flow cytometry. Presence of >1% human CD45^+^ human ALL cells was the threshold to initiate treatment with irinotecan (40 mg/kg). Mice were randomly allocated to the different treatment arms: For the VU9815 PDX model there was a vehicle control (*n* = 3) and irinotecan remission/relapse group (*n* = 4). The UPID 788 PDX model only had an irinotecan remission/relapse group (*n* = 8). Administration occurred i.p. three times per week, for a total of 10 dosages, followed by a 4-week treatment stop. Changes in human leukemic cell burden were monitored every other week. Four mice were excluded from the analysis, as they died immediately after treatment initiation, possibly due to acute tumor lysis syndrome. (**B**) At the end of treatment, mice were euthanized and human leukemic cell infiltration in bone marrow (BM), PB and spleen measured by multi-color flow cytometry. (**C**) Splenomegaly was illustrated by spleen weights and sizes.

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
