# Peer review of "Irinotecan Induces Disease Remission in Xenograft Mouse Models of Pediatric MLL-Rearranged Acute Lymphoblastic Leukemia"

_biomedicines, 2021, doi:10.3390/biomedicines9070711_

Round 1
Reviewer 1 Report
Dear Author,
I find this paper very organized with nice study designs and the idea to use drug repurposing approach in an effort to expedite therapeutics transitioning to clinical is really laudable.
Please address the following questions/comments
1) Please check for typos.. For example in the abstract, line 18 and 19 spell check for camptothecin
2) Fig 1: you have shown cell viability heat map ranging from 0 to 150. What’s your threshold value here to determine whether cell is viable or not. For example, anything that falls over 100 is categorized as viable ?
3) Page 3 of 16: Line 138 and 141. In addition to mentioning that compounds are effective on only one single cell line, please mention the cell line name so the readers can relate it to figures easily.
4) Fig 2: missing label “d”.
5) Page 4 of 16, Line 188. Did you mean 13.9 ?
6) Page 5 of 16. Line 195/196. While your statement holds true i.e. reduced cell numbers after 24 hrs for 25 nm. But the 5nm conc is not exactly the same going by the Fig 3A. Can you comment on that?
Author Response
Dear Author,
I find this paper very organized with nice study designs and the idea to use drug repurposing approach in an effort to expedite therapeutics transitioning to clinical is really laudable.
We would like to thank the reviewer for these compliments and thorough review of our manuscript.
Please address the following questions/comments
1) Please check for typos.. For example in the abstract, line 18 and 19 spell check for camptothecin
We sincerely apologize for the typos in our manuscript and thank the reviewer for pointing this out. The revised version of our manuscript has now thoroughly been screened for typos and corrected where necessary.
2) Fig 1: you have shown cell viability heat map ranging from 0 to 150. What’s your threshold value here to determine whether cell is viable or not. For example, anything that falls over 100 is categorized as viable ?
As the reviewer rightfully assumes, viability percentages of 100% (and above) are deemed completely viable: this has now been underlined in the ‘Materials and Methods’ section of the revised manuscript, line 96 and 100. Intuitively one would expect cell viability to range from 0% to 100%. The viability measures in drug-exposed cell populations, however, are not absolute, but are corrected against untreated (vehicle only) control cells. The number of surviving cells after 4-day cultures in these controls are set to 100% and compared against the viability of drug-treated cells. In rare occasions, mostly at low drug concentrations, leukemic cells tend to initially be modestly stimulated by certain drugs before (at higher drug concentrations) cell viability is notably affected: this sometimes leads to viability measures that exceed 100% when compared to untreated controls. To allow sporadic instances in which this occurs to be presented as ‘viable’ or ‘unaffected by the drug’, the cell viability range was set to range from 0% to 150%.
3) Page 3 of 16: Line 138 and 141. In addition to mentioning that compounds are effective on only one single cell line, please mention the cell line name so the readers can relate it to figures easily.
As suggested by the reviewer, we now mention the name the cell line(s) that we referred to: Line 144, 147-148.
4) Fig 2: missing label “d”.
We thank the reviewer for pointing this out and added the “d” in Figure 2.
5) Page 4 of 16, Line 188. Did you mean 13.9 ?
We thank the reviewer for noticing this, “13,9” should indeed have read 13.9. This has now been corrected in the revised manuscript.
6) Page 5 of 16. Line 195/196. While your statement holds true i.e. reduced cell numbers after 24 hrs for 25 nm. But the 5nm conc is not exactly the same going by the Fig 3A. Can you comment on that?
In all honesty, we are not completely sure what the reviewer means. Fig 3A. shows that for 25 nM the cell counts are clearly lower as compared to the vehicle-treated cells already at 24 hours. The cell counts for the 5 nM treatment at 24 hours are, as expected, not as low as for the 25 nM concentration, but are already lower when compared to the vehicle control. At 48 hours this difference had clearly progressed, and in our opinion, the Annexin+/7-AAD+/- staining in Fig 3B. nicely follow the trend seen for the cell numbers in Fig 3A.Reviewer 2 Report
In this manuscript, Kerstjens, Castro and colleagues demonstrated the efficacy of irinotecan (the prodrug of SN-38) in the treatment of pediatric MLL-rearranged ALL cell lines and PDXs. Overall, the experiments are well designed and executed and the manuscript is well written. The present findings are interesting, yet there are few issues needed to be addressed before accepting this manuscript.
Major points:
1. In Figure. 4D, the representative images of spleen of treatment follow up and WT are unclear.
2. Toxicity concerns: In the murine model of advanced MLL-rearranged ALL, the authors mentioned that one out of the four treated xenotransplanted mice (25%) died on day 26 with no known cause. Did the authors investigate whether this might be related to the toxicity of the used doses of irinotecan? Did the authors monitor/assess body weight changes and CBC for instance ?
Minor points:
1. Few typos have been noticed.
Author Response
In this manuscript, Kerstjens, Castro and colleagues demonstrated the efficacy of irinotecan (the prodrug of SN-38) in the treatment of pediatric MLL-rearranged ALL cell lines and PDXs. Overall, the experiments are well designed and executed and the manuscript is well written. The present findings are interesting, yet there are few issues needed to be addressed before accepting this manuscript.
We thank the reviewer for these kind words and through review of our manuscript.
Major points:
1. In Figure. 4D, the representative images of spleen of treatment follow up and WT are unclear.
We agree with the reviewer that the images of the spleen of the treatment follow-up and WT mice are less clear when compared to the spleen images shown for the vehicle and irinotecan-treated mice. Unfortunately, we do not have better images of these spleens. However, despite of the somewhat lesser quality of these images, we do feel that the images show that the spleens of irinotecan-treated, treatment follow-up and WT mice are comparable in size. We therefore feel that this, in addition to the spleen weight measurements in Fig 4E. and the lack of huCD19+ cells as shown in Fig 4F., convincingly demonstrates that these spleens non-leukemic.
2. Toxicity concerns: In the murine model of advanced MLL-rearranged ALL, the authors mentioned that one out of the four treated xenotransplanted mice (25%) died on day 26 with no known cause. Did the authors investigate whether this might be related to the toxicity of the used doses of irinotecan? Did the authors monitor/assess body weight changes and CBC for instance ?
We thank the reviewer for this important question. To the best of our knowledge this mouse died of an unknown cause: we did not come across notable organ damage or weight loss in any of the irinotecan-treated mice (with or without leukemia). We therefore do not think that irinotecan toxicity is an issue. Nonetheless, we decided to keep this mouse in our study as, like the other mice, it did show a reduction in leukemia burden after the first week of irinotecan treatment (Fig 5A).
Minor points:
1. Few typos have been noticed.
We sincerely apologize for the typos in our manuscript and thank the reviewer for pointing this out. The revised version of our manuscript has now thoroughly been screened for typos and corrected where necessary.